# Increased risk of falls and fractures in patients with psychosis and Parkinson disease

**Joan Forns** [1]*, **J. Bradley Layton** [2], **Jennifer Bartsch**[3], **Mary Ellen Turner**[4], **Colleen Dempsey**[4], **Mary Anthony**[2], **Mary E. Ritchey**[2], **George Demos**[4]

**1** Pharmacoepidemiology and Risk Management, RTI Health Solutions, Barcelona, Spain, **2** Pharmacoepidemiology and Risk Management, RTI Health Solutions, Research Triangle Park, North Carolina, United States of America, **3** Biometrics, RTI Health Solutions, Research Triangle Park, North Carolina, United States of America, **4** ACADIA Pharmaceuticals Inc., San Diego, California, United States of America

* jforns@rti.org

**Data Availability Statement:** The data were obtained pursuant to a license from a third party (IBM Watson Health). In order to provide access to the source study data, the requestor would likely

## Abstract

### Objective

Evaluate whether the risk of falls and fractures differs between patients with Parkinson disease with psychosis (PDP) and patients with Parkinson disease (PD) without psychosis at similar disease stages.

### Methods

Patients with PD without psychosis were identified in the Medicare claims databases (2008–2018) and followed from the first PD diagnosis date during the study period. Patients with a subsequent diagnosis of psychosis were included in the PDP group. Patients with PDP and PD without psychosis were propensity score-matched based on characteristics within blocks of time since cohort entry. The incidence rates (IRs), expressed per 100 person-years, and 95% confidence intervals (CIs) of falls and fractures were evaluated as composite and separate outcomes. Incidence rate ratios (IRRs) were used to compare patients with PDP and PD without psychosis in the matched cohort.

### Results

154,306 patients had PD without psychosis and no falls or fractures before cohort entry; the IR for falls and fractures was 11.41 events (95% CI, 11.29–11.53). 12,127 patients (7.8%) had a subsequent PDP diagnosis. PDP patients had a higher prevalence of most comorbidities and risk factors for falls and fractures than those without psychosis. The crude IR for falls and fractures among PDP patients was 29.03 events (95% CI, 28.27–29.81). PD without psychosis and PDP groups had more falls than fractures. After matching, 24,144 PD patients without psychosis (15.6%) and 12,077 PDP patients (99.6%) were retained. Matched PDP patients had a higher incidence of falls and fractures than PD patients without psychosis (IRR = 1.44; 95% CI, 1.39–1.49). The higher increased rate was noted separately for falls (IRR = 1.48; 95% CI, 1.43–1.54) and any fractures (IRR = 1.17; 95% CI, 1.08–1.27) as well as within specific types of fracture, including pelvis and hip fractures.

need to execute an agreement from IBM Watson Health (https://www.ibm.com/products/marketscan-research-databases/purchase). We confirm that the authors did not have any special access privileges that others would not have.

**Funding:** This study was funded by ACADIA Pharmaceuticals Inc. The authors JF, JBL, JB, MEB, and MA are full-time salaried employees of RTI Health Solutions. MET, CD, and GD are full-time salaried employees of ACADIA. The specific roles of these authors are articulated in the "author contributions" section. Except as disclosed above, the funders otherwise had no role in the study design, data collection and analysis, decision to publish, or preparation of the manuscript.

**Competing interests:** This study was funded by ACADIA Pharmaceuticals Inc. The authors JF, JBL, JB, MEB, and MA are full-time salaried employees of RTI Health Solutions. MET, CD, and GD are full-time salaried employees of ACADIA. Affiliations and funding do not alter our adherence to PLOS ONE policies on sharing data and materials. This manuscript is the result of a study conducted by RTI Health Solutions, sponsored by ACADIA Pharmaceuticals, the maker of pimavanserin (an antipsychotic approved by the US Food and Drug Administration for the treatment of patients with PDP). However, there are no patents, products in development or marketed products associated with this research to declare.

## Conclusions

Our findings suggest a modest but consistently higher increased risk of falls and fractures in PDP patients compared with PD patients without psychosis.

## Introduction

Psychosis is a common complication of Parkinson disease (PD), particularly at advanced stages of the disease, with a prevalence reported to be as high as approximately 75% [1]. Both PD and PD with psychosis (PDP) have been implicated as risk factors for falls and fractures due to several PD-specific factors, including cognitive and motor impairment, postural instability, unsteady gait, bradykinesia, rigidity, increased PD severity, duration of disease, medication use (including dopamine agonists), and frailty [2, 3]. However, it has not yet been established whether the risk of falls and fractures in PD patients differs between those with psychosis and those without.

In patients with PD, the risk of falls is high; approximately 61% of patients report at least one fall and 39% report recurrent falls in study periods ranging between 6 and 29 months [2]. Moreover, the risk of falls and fractures is higher among patients with PD compared with patients without PD [4, 5]. The only study analyzing whether the frequency of falls and fractures differs between patients with PD and those with PDP concluded that the cumulative incidence of falls and fractures was higher in patients with PDP than in patients with PD without psychosis [6]. However, some methodological aspects are lacking in this publication, including specific information about the database and the definitions of falls and fractures. Moreover, a comparative analysis of the risk of falls in both groups of patients is also lacking.

Because of the paucity of evidence in the literature and to better understand the relationship between PDP and risk of falls and fractures, we conducted an observational study in a large, United States (US)–based insurance claims database. The aims of the present study were to understand whether the risk of falls and fractures differed between patients with PDP and patients who have PD without psychosis at similar disease stage and to estimate the absolute risks of falls and fractures among patients with PDP and among those with PD without psychosis.

## Methods

### Data source

This study was conducted using the MarketScan (IBM Watson Health) Commercial Claims and Encounters and the Medicare Supplemental and Coordination of Benefits databases. These commercial insurance databases contain insurance billing data for employees, retirees, and their spouses and dependents with employer-based commercial insurance from approximately 100 large employers across the US. The databases contain information on insurance enrollment, inpatient and outpatient medical procedures and diagnoses, and outpatient pharmacy dispensing of medications.

### Study population

The study population consisted of all adults in MarketScan aged 40 years or older with a recorded diagnosis of PD occurring between January 1, 2008, to June 30, 2018, with at least 6 months of continuous enrollment prior to PD diagnosis date, although gaps in enrollment ≤ 7

days were permitted. Diagnoses were identified with *International Classification of Diseases*, *Ninth Revision*, *Clinical Modification* (ICD-9-CM) and *International Classification of Diseases*, *Tenth Revision*, *Clinical Modification* (ICD-10-CM) diagnosis codes for PD (ICD-9-CM 332.0; ICD-10-CM G20). Patients needed to meet one or more of the following criteria to have a PD diagnosis: (1) one inpatient claim for PD in any recorded diagnosis position; (2) two outpatient claims for PD in any recorded diagnosis position, separated by at least 30 days but within 365 days; or (3) one outpatient claim for PD in any recorded diagnosis position and at least two prescription claims for a PD-related medication (levodopa-carbidopa, anticholinergics, dopamine agonists, monoamine oxidase B inhibitors, or catechol-O-methyltransferase inhibitors) within the 6 months before or after PD diagnosis [5]. For criteria in which more than one code was required, the date of the latest occurring claim was assigned as the PD diagnosis date to avoid immortal person-time (time during which a patient contributes to follow-up but has not yet met all eligibility criteria, and therefore any fall or fracture would be ineligible to be counted as an outcome). Multiple qualifying claims could occur on the same date (e.g., a diagnosis and medication claim could occur on the same date). Patients were excluded if one of the following exclusion criteria occurred at any point prior to the qualifying PD diagnosis date: diagnosis of psychosis (to ensure identification of incident PDP after the PD cohort eligibility date); dispensing of an atypical antipsychotic prescription or haloperidol; diagnosis of bipolar disorder, schizophrenic disorders, or Huntington disease, which are usually treated with antipsychotics; diagnosis of secondary PD, including drug-induced PD, vascular PD, or essential tremor and dementia; or diagnosis of a pathologic fracture that may have resulted from conditions such as cancer, infection, osteomalacia, and Paget's disease. The PD cohort eligibility date was assigned as the date of an individual's first recorded PD diagnosis meeting all the eligibility criteria (see S1 Fig). With application of the exclusion criteria, we assumed that eligible patients with PD in this study were psychosis-free prior to the PD cohort eligibility date, and their person-time (the time, in years, contributed by a patient to the total follow-up time of a cohort) was assigned to the PD without psychosis group starting on their PD cohort eligibility date.

Psychosis was identified on or after the PD cohort eligibility date by using diagnosis codes for conditions related to delusion, hallucinations, psychosis, or paranoia in the inpatient or outpatient setting in any diagnosis position. Antipsychotic medications were not utilized as part of the psychosis definition since there is substantial off-label usage of antipsychotic medications among older individuals, including those with dementia or residents of long-term care facilities without documented psychosis diagnoses [7–9], suggesting that use of an antipsychotic medication may not always indicate actual psychosis symptoms in this population. If a psychosis diagnosis was identified, the patient was censored from the non-PDP group on the date before the psychosis diagnosis; this date was assigned as the PDP index date, and the patient contributed person-time to the PDP group from that point forward (see S2 Fig). Patients could receive a diagnosis of psychosis on the PD cohort eligibility date, and in this instance, they would not contribute any person-time to the PD group.

The same exclusion criteria applied at the time of PD cohort eligibility date, except the psychosis diagnosis, were applied again at the date of the psychosis diagnosis by using all available claims prior to the PDP index date.

## Study outcomes

Recurrent falls and fractures were assessed as outcomes in the following categories: all falls and fractures (composite falls/fractures); falls only; fractures only; and site-specific fractures of key

interest (femur fracture, hip fracture, pelvis fracture, upper-limb fracture, and vertebral fracture).

Falls were identified by using both inpatient and outpatient ICD-9-CM and ICD-10-CM diagnosis codes occurring in any recorded diagnosis position. Repeated codes for falls occurring within 7 days were considered part of the same event, and patients did not contribute at-risk person-time during this 7-day period [5]; the date of the first diagnosis during this period was assigned as the fall's event date. Occurrence of falls was additionally evaluated in the 30 days before cohort eligibility to distinguish between new falls that occurred during follow-up from continuing care and falls that occurred prior to the beginning of follow-up.

Fractures were similarly defined by using inpatient and outpatient ICD-9-CM and ICD-10-CM diagnosis codes in any diagnostic position, and each fracture diagnosis was categorized by body site (skull, vertebrae, trunk, upper limb, hand and wrist, pelvis, hip, femur, lower leg, foot, and ankle). In order to ensure identification of a new fracture, the fracture diagnosis code was required to be paired with a procedure code (ICD-9-CM Procedure, ICD-10 Procedure Code System, Current Procedural Terminology [CPT], or Healthcare Common Procedure Coding System [HCPCS] codes) for a site-specific fracture repair occurring in the 7 days before or after diagnosis. If the diagnosis and the repair code occurred on separate dates, the earlier of the two dates was assigned as the fracture event date. All subsequent fractures that occurred at the same site within 1 year after the initial site-specific fracture were considered part of the same event, and individuals did not contribute at-risk person-time for a same-site recurrent fracture during that 1-year period, consistent with previous study of fractures in patients with PD [5]. The occurrence of fractures up to a year prior to the beginning of follow-up was evaluated by using all prior available claims data to account for recurrent, site-specific fractures. Fractures were evaluated across all sites combined and separately as site-specific outcomes for major sites (hip, pelvis, femur, vertebrae, upper limb). For the any-fracture outcome, site-specific fractures at different sites occurring within 7 days of each other were considered part of the same event; the date of the earlier-occurring fracture was assigned as the any-fracture event date, and the patient was not considered at risk for 7 days after that date. For the composite falls/fractures outcome, falls and fractures occurring on the same date or within 7 days were considered part of the same event and were counted only once.

## Covariates

Demographic and clinical characteristics prior to the beginning of follow-up were evaluated for use in descriptive analyses and as covariates in the propensity score models. Among the demographic characteristics, we included age, sex, year, and frailty indicators (such as wheelchair use, ambulance/life support, bladder dysfunction, coagulopathy, home oxygen, paralysis, dementia, cancer screening, heart failure, lipid abnormality, vertigo, difficulty walking, podiatric care, rehabilitation services, arthritis, skin ulcer, sepsis, stroke/brain injury, weakness, diabetes mellitus complications, home hospital bed) [10, 11]. In addition, we collected clinical characteristics, including additional components of the Charlson Comorbidity Index (e.g., myocardial infarction, peripheral vascular disease, or chronic obstructive pulmonary disease), additional predictors of falls or fractures (e.g., delirium, osteoporosis, or multiple sclerosis), concomitant medications assessed using a 6- to 12-month look-back period (if the patient had more than 6 months of baseline data available), PD drugs, and measures of health care utilization (number of hospitalizations, number of emergency department visits).

## Follow-up

Patients were followed from the analysis-specific start of follow-up, and they were censored at the first occurrence of the following: end of study time period (June 30, 2018); disenrollment from the MarketScan databases; occurrence of pathological fracture that may have resulted from conditions such as cancer, infection, osteomalacia, and Paget's disease; or diagnosis of bipolar disorder, schizophrenic disorders, or Huntington disease. Patients in the PD without psychosis group were censored if they received a diagnosis of new-onset psychosis, at which point they were switched to the PDP group.

## Statistical analysis

Descriptive statistics were used to compare baseline characteristics between the groups of patients with and without psychosis, both in the unmatched and matched cohorts. Counts and proportions of patients with each covariate were presented for categorical variables, while continuous variables were summarized using means and standard deviations (SDs). Standardized mean differences (SMDs) were used to quantify imbalances in baseline covariate distributions between the PD without psychosis and PDP groups [12]. The SMD was calculated as the difference in means or proportions, divided by the pooled SDs [12].

To more directly compare the rates of falls and fractures between patients with PDP and those with PD without psychosis, we identified a cohort of patients with PDP matched to patients with PD without psychosis at similar disease trajectories based on time since cohort entry and relevant clinical claims using a sequential propensity score-matching approach (See S1 Methods). Briefly, within each consecutive 4-month block, a multivariable logistic regression model was used to estimate block-specific propensity scores using the covariates described in previous sections. After the block-specific propensity scores were estimated, PD without psychosis index dates were matched to the PDP index dates with a 2:1 fixed-ratio matching using a greedy nearest neighbor 5- to 1-digit matching algorithm, without replacement [13], with a maximum caliper of 0.2 times the SD of the estimated logit of the propensity score [14].

Within the unmatched and propensity-matched cohorts, we estimated crude incidence rates (IRs) for the composite falls/fracture, falls, and fractures as the number divided by the duration of follow-up, as well as accompanying 95% confidence intervals (CIs) [15]. For comparisons of the IRs across PD without psychosis and PDP groups in the matched cohorts, the incidence rate ratios (IRRs) and 95% CIs were estimated by dividing the group-specific IRs and estimating the standard errors [16].

IRRs were estimated across the entire follow-up period, for each year increment of follow-up (0–1 year, > 1–2 years, > 2–3 years, > 3 years– 4 years, > 4 years), and for subgroups of age at the index date (40 to < 65 years, 65 to < 70 years, 70 to < 85 years, ≥ 85 years). All analyses were performed using SAS 9.4 (SAS Institute, Inc., Cary, North Carolina).

Finally, due to the possibility of PD diagnostic misclassification, we repeated the main analysis but excluded patients with a diagnosis code for atypical PD (i.e., dementia with Lewy body, multiple system atrophy, or progressive supranuclear palsy) at baseline.

## Results

We identified 154,841 patients with PD; 154,339 of these patients contributed person-time to the PD without psychosis group, but 502 did not contribute time to the PD without psychosis group because they received a diagnosis of PDP on the PD cohort eligibility date. Among patients with PD, 12,132 received a diagnosis of psychosis on or after the cohort eligibility date and were subsequently included in the PDP group (Fig 1).

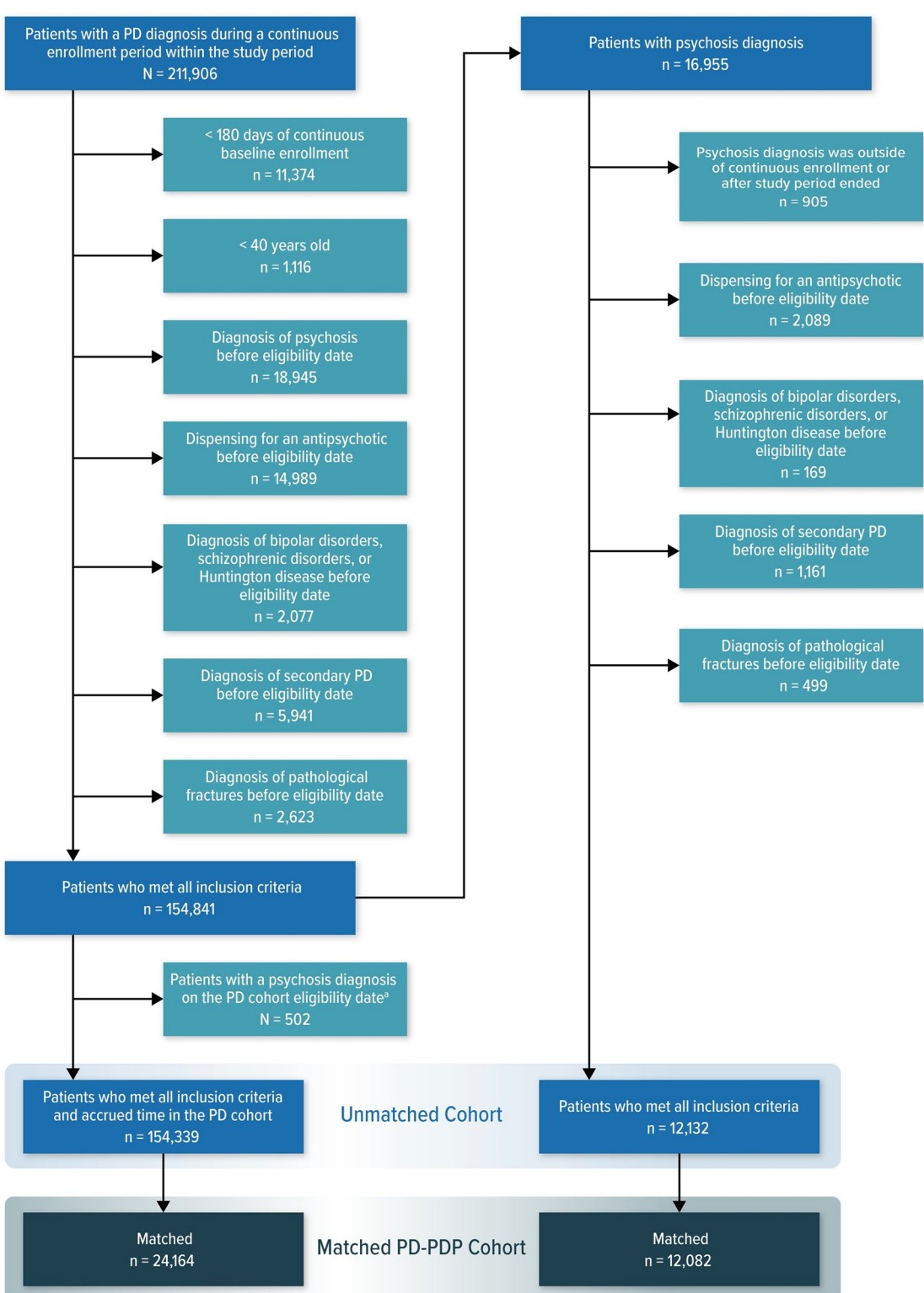

**Fig 1. Attrition of the Parkinson disease cohort by application of eligibility criteria.** PD = Parkinson disease; PDP = Parkinson disease with psychosis. [a] Those 502 patients with a PD diagnosis and all eligibility criteria could not contribute time to the PD without psychosis group because they were diagnosed with psychosis on the cohort eligibility date.

## Characteristics of the unmatched patients with PDP and those with PD without psychosis

Characteristics of the unmatched PD without psychosis and PDP groups are shown in Table 1. Patients in the PDP group were generally sicker than those in the PD without psychosis group, and they had higher rates of comorbidities than the PD without psychosis group, including arthritis (70% vs. 46%), lipid abnormalities (62% vs. 48%), heart failure (35% vs. 16%), and dementia (61% vs. 27%). Occurrence of more serious events and comorbidities was much higher in the PDP group as compared with the PD without psychosis group, including ambulance/life support (58% vs. 21%) and sepsis (35% vs. 16%). Compared to patients with PD without psychosis, higher proportions of patients with PDP had risk factors associated with falls and fractures, including difficulty walking (58% vs. 29%), vertigo (33% vs. 17%), delirium (50% vs. 14%), weakness (36% vs. 14%), malnutrition (21% vs. 8%), and osteoporosis (16% vs. 9%).

## Crude IRs for falls and fractures in unmatched patients with PDP and PD without psychosis

Crude incidence of falls and fractures among patients with PDP was higher than that among patients with PD without psychosis (Table 2 and S1 Table). The IR of the composite falls/fractures outcome was 29.03 per 100 person-years (95% CI, 28.27–29.81) in the PDP group, whereas it was 11.41 events per 100 person-years (95% CI, 11.29–11.53) in the PD without psychosis group. More of the outcome events were falls than fractures. There were differing time trends in the rates of occurrence of falls and fractures between groups. In the PD without psychosis group, the IR generally increased over the 4 years of follow-up after the PD cohort eligibility date, but in the PDP group, the highest IR of falls and fractures occurred in the first year following the psychosis diagnosis (Table 2 and S1 Table). Additionally, in both groups, there were trends for increasing fall and fracture incidence with increasing age, although in the PDP group, the increasing risks plateaued somewhat at older ages, with little difference observed between those aged 70 to < 85 years and those aged ≥ 85 years (Table 2 and S2 Table).

## Comparative analysis between matched PD without psychosis and PDP groups

Of the 12,132 patients with PDP in the unmatched cohort, 12,082 (99.6%) successfully matched to similar patients in the PD without psychosis group. A 2:1 matching ratio was implemented, so of the original 154,339 patients with PD without psychosis, 24,164 (15.7%) were retained by matching (Fig 1). The characteristics of the matched patients were well-balanced between the PD without psychosis and PDP groups, as evidenced by the SMDs for all characteristics being near 0, between –0.04 and 0.03 (S3 Table).

There were 5,434 fall or fracture events identified in the matched PDP group and 7,497 in the matched PD without psychosis comparator group (Table 3). Within the matched groups, patients with PDP had higher IRs of the composite falls/fractures outcome (IRR = 1.44; 95% CI, 1.39–1.49), falls (IRR = 1.48; 95% CI, 1.43–1.54), and fractures (IRR = 1.17; 95% CI, 1.08–1.27) (Table 3). The IRR estimates had narrow confidence intervals, due to the large number of patients and outcomes in these databases. The vast majority of the composite falls/fractures events were falls, with a relatively smaller number of fracture events. When investigating the association of PDP with five specific fracture sites of key interest, the number of cases was small, except for hip fractures, which resulted in a modest elevated risk (IRR = 1.17; 95% CI,

**Table 1. Descriptive characteristics (at least in 10% of patients) of patients with PDP and patients with PD without psychosis, before matching.**

| Demographic | Overall[a] (N = 154,841) | PD without psychosis[b] (n = 154,339) | PDP[c] (n = 12,132) |
|---|---|---|---|
| Age at index, mean (SD) | 72.2 (11.56) | 72.2 (11.56) | 78.1 (9.24) |
| Sex, female, n (%) | 61,658 (39.8) | 61,472 (39.8) | 4,773 (39.3) |
| Frailty indicators, n (%) | | | |
| Ambulance/life support | 33,006 (21.3) | 32,820 (21.3) | 7,073 (58.3) |
| Arthritis | 71,362 (46.1) | 71,103 (46.1) | 8,427 (69.5) |
| Bladder dysfunction | 23,040 (14.9) | 22,940 (14.9) | 4,038 (33.3) |
| Cancer screening | 42,339 (27.3) | 42,244 (27.4) | 3,570 (29.4) |
| Dementia | 41,702 (26.9) | 41,463 (26.9) | 7,416 (61.1) |
| Difficulty walking | 44,696 (28.9) | 44,514 (28.8) | 7,067 (58.3) |
| Heart failure | 24,723 (16.0) | 24,604 (15.9) | 4,201 (34.6) |
| Lipid abnormality | 73,850 (47.7) | 73,598 (47.7) | 7,461 (61.5) |
| Sepsis | 24,953 (16.1) | 24,854 (16.1) | 4,229 (34.9) |
| Stroke/brain injury | 20,013 (12.9) | 19,901 (12.9) | 4,015 (33.1) |
| Vertigo | 26,662 (17.2) | 26,529 (17.2) | 4,038 (33.3) |
| Weakness | 21,935 (14.2) | 21,824 (14.1) | 4,346 (35.8) |
| Components of Charlson Comorbidity Index, n (%) | | | |
| Chronic obstructive pulmonary disease | 25,905 (16.7) | 25,796 (16.7) | 3,481 (28.7) |
| Diabetes mellitus | 37,095 (24.0) | 36,947 (23.9) | 3,971 (32.7) |
| Peripheral vascular disease | 26,450 (17.1) | 26,318 (17.1) | 4,179 (34.4) |
| Tumor | 20,755 (13.4) | 20,678 (13.4) | 2,486 (20.5) |
| Other risk factors of falls or fractures, n (%) | | | |
| Delirium | 22,233 (14.4) | 22,061 (14.3) | 6,031 (49.7) |
| Depression | 25,705 (16.6) | 25,584 (16.6) | 4,211 (34.7) |
| Malnutrition | 11,980 (7.7) | 11,910 (7.7) | 2,545 (21.0) |
| Orthostatic hypotension | 5,431 (3.5) | 5,399 (3.5) | 1,482 (12.2) |
| Osteoporosis | 13,835 (8.9) | 13,776 (8.9) | 1,886 (15.5) |
| Concomitant comedications[d], n (%) | | | |
| Anticholinesterase inhibitors | 11,821 (7.6) | 11,762 (7.6) | 2,084 (17.2) |
| Antidepressants | 37,076 (23.9) | 36,965 (24.0) | 3,871 (31.9) |
| Benzodiazepines | 24,398 (15.8) | 24,327 (15.8) | 2,394 (19.7) |
| Diuretics | 34,042 (22.0) | 33,943 (22.0) | 3,124 (25.8) |
| PD drugs[e] | 80,564 (52.0) | 80,441 (52.1) | 6,672 (55.0) |
| Health care utilization, mean (SD)[f] | | | |
| Number of hospitalizations | 0.2 (0.47) | 0.2 (0.47) | 0.5 (0.72) |
| Number of emergency department visits | 0.5 (1.07) | 0.5 (1.07) | 1.3 (1.69) |

PD = Parkinson disease; PDP = Parkinson disease with psychosis; SD = standard deviation.

Note: All characteristics were assessed during the entire look-back period unless otherwise stated.

[a] All patients were evaluated at their PD cohort eligibility date.

[b] Patients who did not have a psychosis diagnosis on or before their PD cohort eligibility date.

[c] Patients who met the criteria to enter the PDP cohort; evaluated at their first psychosis diagnosis date.

[d] Assessed in a look-back period of up to 1 year before the corresponding cohort entry/eligibility date.

[e] Comprised levodopa-carbidopa, anticholinergics, dopamine agonists, monoamine oxidase B inhibitors, and catechol-O-methyltransferase inhibitors.

[f] Assessed in the 6 months before the corresponding cohort entry/eligibility date.

1.04–1.32). For the rest of fractures, the small number of cases resulted in IRRs estimates with wide confidence intervals, although the estimate for pelvis fractures was elevated (IRR = 1.57; 95% CI, 0.81–2.99), while the IRRs for femur and upper-limb fractures were closer to null; the IRR for vertebrae fractures was below the null (Table 3).

**Table 2. Crude incidence rates of composite falls/fractures for the unmatched PD without psychosis[a] and PDP[b] groups overall, by time interval and by age groups.**

| Outcome in Specified Time Period | Group | No. of patients | No. of events | No. of person-years | IR (95% CI)[c] per 100 person-years |
|---|---|---|---|---|---|
| Overall | PDP | 12,127 | 5,453 | 18,783 | 29.03 (28.27–29.81) |
| | PD[a] | 154,306 | 36,341 | 318,488 | 11.41 (11.29–11.53) |
| Time interval, year | | | | | |
| 0–1 | PDP | 12,127 | 2,925 | 8,587 | 34.06 (32.84–35.32) |
| | PD[a] | 154,306 | 13,471 | 120,925 | 11.14 (10.95–11.33) |
| > 1–2 | PDP | 6,275 | 1,088 | 4,784 | 22.74 (21.41–24.14) |
| | PD[a] | 93,663 | 7,758 | 75,665 | 10.25 (10.03–10.48) |
| > 2–3 | PDP | 3,571 | 671 | 2,723 | 24.64 (22.81–26.58) |
| | PD[a] | 59,346 | 5,055 | 48,390 | 10.45 (10.16–10.74) |
| > 3–4 | PDP | 2,010 | 403 | 1,462 | 27.57 (24.95–30.40) |
| | PD[a] | 38,293 | 3,541 | 30,653 | 11.55 (11.17–11.94) |
| > 4 | PDP | 1,014 | 366 | 1,228 | 29.81 (26.83–33.02) |
| | PD[a] | 23,980 | 6,516 | 42,854 | 15.21 (14.84–15.58) |
| Age group, years | | | | | |
| 40 to < 65 | PDP | 1,206 | 333 | 1,856 | 17.94 (16.06–19.97) |
| | PD[a] | 45,676 | 5,083 | 97,978 | 5.19 (5.05–5.33) |
| 65 to < 70 | PDP | 839 | 357 | 1,603 | 22.27 (20.02–24.71) |
| | PD[a] | 15,436 | 3,354 | 36,012 | 9.31 (9.00–9.63) |
| 70 to < 85 | PDP | 6,893 | 3,482 | 11,279 | 30.87 (29.85–31.91) |
| | PD[a] | 70,279 | 20,962 | 148,917 | 14.08 (13.89–14.27) |
| ≥ 85 | PDP | 3,189 | 1,281 | 4,045 | 31.67 (29.96–33.45) |
| | PD[a] | 22,915 | 6,942 | 35,580 | 19.51 (19.05–19.98) |

CI = confidence interval; IR = incidence rate; PD = Parkinson disease; PDP = Parkinson disease with psychosis.

[a]All patients with PD who did not have a first psychosis diagnosis on their PD eligibility date.

[b]Patients who met the criteria to enter the PDP cohort.

[c]The number of events was assumed to follow a Poisson distribution; corresponding exact 95% CIs were computed using methods described in Dobson et al. [15].

When stratified by follow-up time, the largest increased IRR was observed in the period 0-1 years from the index date for all outcomes (Fig 2), consistent with the increased IR observed in the PDP group in the crude analyses. While the IRRs observed in later periods of follow-up for the composite falls/fractures outcome and falls alone were lower than the 0- to 1-year interval, they were still consistently increased above the null. For the any-fracture outcome, the IRR was highest in the 0- to 1-year interval, but it was attenuated to the null during the later periods of follow-up.

## Sensitivity analysis: Comparative analysis between matched PD without psychosis and PDP groups excluding patients with atypical PD at baseline

A total of 2,678 (11.1%) and 1,385 (11.5%) patients in the matched PD and PDP cohorts were excluded due to the co-occurrence of a diagnosis code for atypical PD (Lewy body dementia or degenerative diseases of the basal ganglia). After those exclusions, the characteristics of the remaining patients in the matched cohort were well-balanced between the PD without psychosis and PDP groups, as evidenced by the SMDs for all characteristics being near 0, between −0.05 and 0.03 (S4 Table). Comparative matched analyses excluding patients with atypical PD were almost identical to those observed in the main analysis (S5 Table); patients with PDP had

**Table 3. Incidence rates and incidence rate ratios of falls and fractures for the matched PD-PDP cohort[a].**

| Outcome | Group | No. of patients | No. of events | No. of person-years | IR (95% CI)[b] per 100 person-years | IRR (95% CI)[c] |
|---|---|---|---|---|---|---|
| Composite falls/fractures | PDP | 12,077 | 5,434 | 18,735 | 29.00 (28.24–29.79) | 1.44 (1.39–1.49) |
| | PD[d] | 24,144 | 7,497 | 37,211 | 20.15 (19.69–20.61) | Reference |
| Falls | PDP | 12,078 | 4,859 | 18,746 | 25.92 (25.20–26.66) | 1.48 (1.43–1.54) |
| | PD[d] | 24,147 | 6,512 | 37,230 | 17.49 (17.07–17.92) | Reference |
| Any fracture | PDP | 12,081 | 941 | 18,823 | 5.00 (4.68–5.33) | 1.17 (1.08–1.27) |
| | PD[d] | 24,156 | 1,597 | 37,324 | 4.28 (4.07–4.49) | Reference |
| Femur | PDP | 12,030 | 147 | 18,682 | 0.79 (0.66–0.92) | 1.15 (0.93–1.41) |
| | PD[d] | 24,084 | 254 | 37,117 | 0.68 (0.60–0.77) | Reference |
| Hip | PDP | 11,932 | 433 | 18,379 | 2.36 (2.14–2.59) | 1.17 (1.04–1.32) |
| | PD[d] | 23,924 | 739 | 36,658 | 2.02 (1.87–2.17) | Reference |
| Pelvis | PDP | 12,080 | 19 | 18,828 | 0.10 (0.06–0.16) | 1.57 (0.81–2.99) |
| | PD[d] | 24,158 | 24 | 37,335 | 0.06 (0.04–0.10) | Reference |
| Upper limb | PDP | 12,031 | 198 | 18,616 | 1.06 (0.92–1.22) | 1.14 (0.96–1.37) |
| | PD[d] | 24,075 | 344 | 36,993 | 0.93 (0.83–1.03) | Reference |
| Vertebrae | PDP | 12,075 | 17 | 18,829 | 0.09 (0.05–0.14) | 0.75 (0.40–1.33) |
| | PD[d] | 24,157 | 45 | 37,327 | 0.12 (0.09–0.16) | Reference |

CI = confidence interval; IR = incidence rate; IRR = incidence rate ratio; PD = Parkinson disease; PDP = Parkinson disease with psychosis.

[a] Patients who met the criteria to enter the PDP cohort were evaluated at their psychosis diagnosis date and were matched; patients with PD without psychosis who were selected for the matched cohort were evaluated at the date of the matched PD diagnosis.

[b] The number of events was assumed to follow a Poisson distribution; corresponding exact 95% CIs were computed using methods described in Dobson et al. [15].

[c] The number of events was assumed to follow a Poisson distribution; corresponding exact 95% CIs were computed using methods described in Sahai and Khurshid [16].

[d] Patients with PD without psychosis.

higher IRRs of the composite falls/fractures outcome (IRR = 1.44; 95% CI, 1.39–1.50), falls (IRR = 1.48; 95% CI, 1.43–1.54), and fractures (IRR = 1.18; 95% CI, 1.08–1.28).

## Discussion

This is one of the first studies attempting to examine whether the risk of falls and fractures differs between patients with PD without psychosis and those with PDP. The results of this study included 154,339 unique patients with PD, 12,132 of whom also received a diagnosis of PDP (24,164 patients with PD without psychosis and 12,082 patients with PDP in the matched analysis) and suggest a modest and consistently increased risk of falls and fractures in patients with PDP compared with patients with PD without psychosis. The observed risk was highest early after the psychosis diagnosis.

Because the risk of falls may be influenced by different factors, including PD severity, duration of the disease, age, or antipsychotic use, we decided a priori to match patients with PDP and those with PD without psychosis at time points where their characteristics were similar. Additionally, we matched patients within time blocks since cohort entry to address selection bias by survival, as patients included in the PDP group were required to survive in the data for some period of time before receiving a diagnosis of psychosis. The decision to match patients was reinforced because patients within our study were older and had more comorbidities, more frailty indicators, and more antipsychotic use at the time of PDP diagnosis than at the time of the initially identified PD diagnosis. A 2:1 matching ratio was successfully implemented, with almost all patients with PDP (99.6%) and 15.7% of patients with PD without

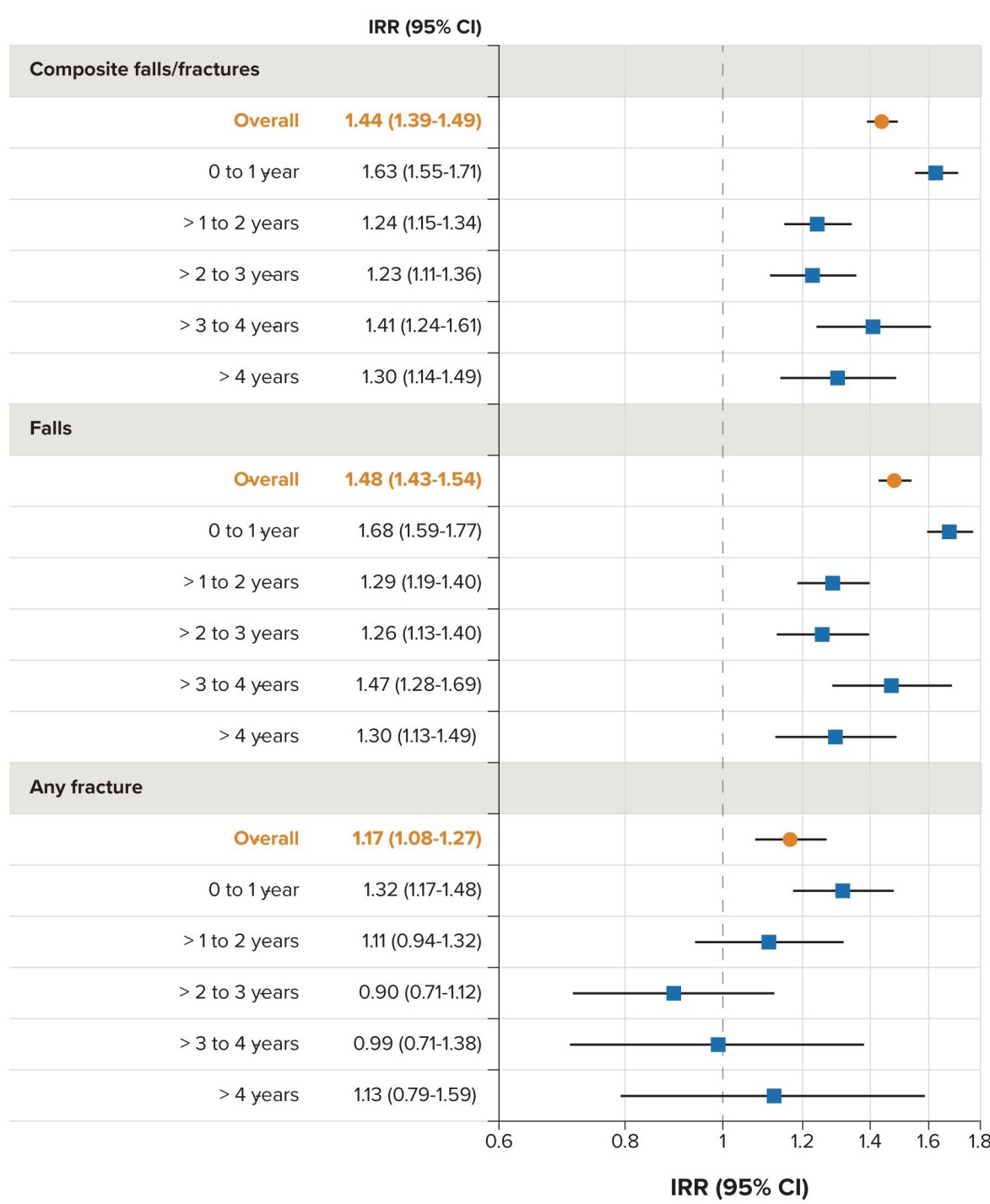

**Fig 2. Incidence rate ratios of falls and fractures for the matched PD-PDP cohort, overall and by time interval.** CI = confidence interval; IRR, incidence rate ratio; matched PD-PDP cohort = cohort of patients with PD without PDP matched to patients with PDP; PD = Parkinson disease; PDP = Parkinson disease with psychosis.

psychosis being retained and included in the analysis. While a large proportion of patients with PD without psychosis were excluded by the matching, the characteristics of the resulting matched cohort were very well-balanced between the PD without psychosis and PDP groups (e.g., mean age of 78 years, greater than 60% proportion of male patients, and a high

prevalence of comorbidities and risk factors for falls and fractures). The risk of falls and fractures was modestly but consistently higher in patients with PDP than in those with PD without psychosis in the matched analyses.

A previous retrospective claims analysis was conducted among 1,066 patients with PDP and 28,250 patients with PD without psychosis. The cumulative incidence across a 12-month time period of "at least 1 fall" was 2.4% (26/1,066) for patients with PDP and 0.7% (198/28,250) for patients with PD without psychosis. The cumulative incidence of "at least 1 fracture" was 16.9% (180/1,066) for patients with PDP and 7.3% (2,062/28,250) for patients with PD without psychosis across a 12-month time period [6]. There are differences between our approach and that of Fredericks et al. [6]; however, the conclusions of Fredericks et al. [6] are aligned with those obtained in the present study, suggesting a higher rate of falls and fractures in patients with PDP than in patients with PD without psychosis, although their finding of a higher rate for fractures than for falls differed from our study. However, since methodological aspects are lacking in this publication (e.g., falls and fracture definition), we cannot elaborate further on this. Finally, the population in Fredericks et al. [6] was younger (mean age for patients with PD and those with PDP, 69.6 and 74.5 years, respectively) compared with our study (mean age, 72.2 and 78.1, respectively).

The increased risk for falls and fractures in patients with PDP compared with patients with PD without psychosis may be attributed to age, disease trajectory, comorbidities, medication use, or a sudden change in disease symptomology. A review of the literature identified disease severity as one of the factors most strongly associated with recurrent falls in patients with PD [2]. In our study, we accounted for these variables in our design and analysis to the extent possible using diagnoses, procedures, and medication dispensing in claims data. After our accounting for other risk factors, psychosis appears to be an independent risk factor to explain the risk of falls. The development of psychosis in patients with PD has been linked to several factors, including the use of dopaminergic drugs to treat PD, older age, later PD onset, higher disease severity, longer duration of PD, and cognitive impairment or depression [17]. Interestingly, the observed increased risk of falls and fractures in our study among patients with PDP was the highest in the 0- to 1-year interval after their having received the diagnosis of psychosis. This increased risk of falls on or during the first year after psychosis diagnosis date might be explained by several reasons. First, the onset of psychosis might be associated with comorbidities such as infection, dehydration, sleep deprivation, irregular nutrition, or psychosocial stress such as hospital admissions that might increase the risk of falls. Once these factors are stabilized, the risk of falls might decrease [18]. Second, some of the pharmacological strategies to treat acute onset of psychosis, including reduction of antiparkinsonian drugs and introduction of antipsychotic drugs, might increase the risk of falls [17].

The results of the present study should be evaluated in view of its potential limitations. As this study was conducted in existing administrative claims data that are generated primarily for billing purposes rather than for clinical diagnoses or research, many of the limitations of the study arise from the use of coded diagnosis and procedure information rather than clinical data. This may introduce the potential for missing or misclassified study variables. This potential for misclassification might affect the study outcome since not all falls may be medically attended; thus, falls identified with diagnosis codes likely represent a subset of all falls, perhaps only more severe events. Similarly, not all fractures may require repair (which was a requirement of our identification algorithm), and thus only a subset of the more severe events would be represented by our case algorithm; or conversely, fractures among frail individuals with limited life expectancy may be less likely to be repaired. This potential misclassification may result in undercounting of true fall or fracture events. In addition, while prescription claims for medications indicate that a prescription has been dispensed by a pharmacy, it may not

reflect actual exposure to medications and use by the patient. Also, inpatient medication history is not available, which may represent a potential misclassification for some measures of medication use.

We attempted to match patients in the PD and PDP cohorts at similar disease trajectories based on time since cohort entry (i.e., first noted PD diagnosis) and relevant clinical claims. Nevertheless, the success of matching patients at similar disease trajectories is unclear due to the lack of granular clinical data in administrative claims data, such as severity of initial symptoms, progression of clinical symptoms, and other key variables. In the same line, the possibility that a patient with transient psychosis has been classified as PDP cannot be rejected. Our eligibility criteria in the PDP cohort only required the presence of a psychosis diagnosis after the PD diagnosis without requiring repeated codes. Claims data may lack the necessary granularity to define duration or intensity of symptoms. The possibility of diagnostic misclassification due to the inclusion of patients with atypical PD cannot be fully rejected. Our strict PD case definition used in the present study reduced the risk of diagnostic misclassification. Moreover, main analyses were repeated excluding patients with atypical PD at baseline. The results of this sensitivity analysis were almost identical as observed in the main analysis.

Most of the previous studies of patients with PDP identified patients with PDP using prescription of antipsychotics instead of using diagnosis codes to identify psychosis. Conversely, in our study, we utilized a claims-based algorithm of psychosis reliant on diagnosis coding alone after the PD diagnosis. This strict criterion was used to identify patients and avoid the inclusion of patients with other symptoms for which antipsychotics can be prescribed, such as delirium, dementia, or agitation. However, the identification of a clinical diagnosis of psychosis might be challenging in an administrative claims database using coded diagnoses, as claims are created for billing and reimbursement rather than the purpose of describing clinical findings. Validation of the psychosis algorithm used in the present study is warranted.

The US transitioned from ICD-9-CM to ICD-10-CM coding systems for diagnoses in 2015. We utilized claims-based definitions of PD and falls or fractures that have been used previously [5], but these definitions were all developed in ICD-9-CM. Although we conducted both forward and backward mapping of the ICD-9-CM to ICD-10-CM codes using the US Centers for Medicare and Medicaid Services General Equivalence Mapping code crosswalks, differences in the conceptual meaning and usage of the codes may exist over time. Additionally, the transition from ICD-9-CM to ICD-10-CM coding was not instantaneous, and some mixtures of coding types may exist at the same time in the data.

Additionally, fundamental differences are present in the characteristics of patients with PDP compared with patients with PD without psychosis (e.g., older patients with more comorbidities and more risk factors for falls). Thus, we attempted to identify patients who were comparable in the PDP and PD without psychosis groups for comparison. While large numbers of covariates were employed in the propensity score models, and the balancing of the measured covariates—including proxies for frailty—was very successful, the possibility of residual unmeasured confounding cannot be totally rejected. Additionally, the secondary analyses of IRs over time utilize slightly different starting times for the PD without psychosis group (in which patients might be matched to the PDP group at a later date than their date of PD cohort eligibility).

In summary, the present study provides estimates of IRs of falls and fractures for 154,339 patients with PD without psychosis and 12,132 with PDP and suggests a modest but consistent increased risk of falls and fractures in patients with PDP compared with patients with PD without psychosis.

## Supporting information

**S1 Fig. Assignment of the cohort eligibility date.** PD = Parkinson disease.
(DOCX)

**S2 Fig. Assignment of person-time to the Parkinson disease with and without psychosis groups relative to the PD cohort eligibility date and the PDP index date.** PD = Parkinson disease; PDP = Parkinson disease with psychosis.
(DOCX)

**S1 Methods. Sequential propensity score matching approach.**
(DOCX)

**S1 Table. Crude incidence rates of falls and fractures for the unmatched PD without psychosis and PDP groups, overall and by time interval.** CI = confidence interval; IR, incidence rate; PD = Parkinson disease; PDP = Parkinson disease with psychosis. [a] All patients with PD who did not have a first psychosis diagnosis on their PD eligibility date. [b] Patients who met the criteria to enter the PDP cohort. [c] The number of events was assumed to follow a Poisson distribution; corresponding exact 95% CIs were computed using methods described in Dobson et al. (1991).
(DOCX)

**S2 Table. Incidence rates of falls and fractures for the unmatched PD without psychosis and PDP groups, by age category at index date.** CI = confidence interval; IR, incidence rate; PD = Parkinson disease; PDP = Parkinson disease with psychosis. [a] All patients with PD who did not have a first psychosis diagnosis on their PD eligibility date. [b] Patients who met the criteria to enter the PDP cohort. [c] The number of events was assumed to follow a Poisson distribution; corresponding exact 95% CIs were computed using methods described in Dobson et al. (1991).
(DOCX)

**S3 Table. Descriptive characteristics of patients with Parkinson's disease with and without psychosis, matched cohort and excluded from matched cohort.** HIV/AIDS = human immunodeficiency virus/acquired immune deficiency syndrome; PD = Parkinson disease; PDP = Parkinson disease with psychosis; SD = standard deviation. Note: All characteristics were assessed during the entire look-back period unless otherwise stated. [a] Patients with PD without psychosis who were selected for the matched cohort; evaluated at the index date of the matched PD diagnosis. [b] Patients who met the criteria to enter the PDP cohort; evaluated at their first psychosis diagnosis date. [c] Patients who did not develop psychosis at the PD cohort eligibility date and were not selected for the matched cohort. Evaluated at their PD cohort eligibility date. [d] Assessed in a look-back period of up to 1 year before the corresponding cohort entry/eligibility date. [e] Comprised all systemic glucocorticoids (excluded nonsystemic administration routes such as topical or inhaled applications). [f] Comprised levodopa-carbidopa, anticholinergics, dopamine agonists, monoamine oxidase B inhibitors, and catechol-O-methyltransferase inhibitors. [g] Assessed in the 6 months before the corresponding cohort entry/eligibility date.
(DOCX)

**S4 Table. Sensitivity analysis of descriptive characteristics of patients with Parkinson's disease with and without psychosis, matched cohort excluding patients with atypical Parkinson's disease.** HIV/AIDS = human immunodeficiency virus/acquired immunodeficiency syndrome; ICD-9-CM = *International Classification of Diseases, Ninth Revision, Clinical*

*Modification*; ICD-10-CM = *International Classification of Diseases*, *Tenth Revision*, *Clinical Modification*; PD = Parkinson's disease; PDP = Parkinson's disease psychosis; SD = standard deviation. Note: All characteristics were assessed during the entire look-back period unless otherwise stated. ICD-9-CM codes to identify atypical PD were 33182 and 3330. ICD-10-CM codes to identify atypical PD were G3183, G903, G239, G238, G232, G230, and G231. [a] Patients with PD who were selected for the matched cohort; evaluated at the index date of the matched PD diagnosis. [b] Patients who met the criteria to enter the PDP cohort; evaluated at their first psychosis diagnosis date. [c] Patients who did not develop psychosis at the PD eligibility date and were not selected for the matched cohort. Evaluated at their PD cohort eligibility date. [d] Assessed in a look-back period of up to 1 year before the corresponding cohort entry/ eligibility date. [e] Comprised all systemic glucocorticoids (excluded nonsystemic administration routes such as topical or inhaled applications). [f] Comprised levodopa-carbidopa, anticholinergics, dopamine agonists, monoamine oxidase B inhibitors, and catechol-O-methyltransferase inhibitors. [g] Assessed in the 6 months before the corresponding cohort entry/eligibility date.
(DOCX)

**S5 Table. Sensitivity analysis of incidence rates and incidence rate ratios of falls and fractures for the matched PD-PDP cohort[a] excluding patients with atypical Parkinson's disease.** CI = confidence interval; ICD-9-CM = International Classification of Diseases, Ninth Revision, Clinical Modification; ICD-10-CM = International Classification of Diseases, Tenth Revision, Clinical Modification; IR = incidence rate; IRR = incidence rate ratio; PD = Parkinson's disease; PDP = Parkinson's disease psychosis; PY = person-year. Note: ICD-9-CM codes to identify atypical PD were 33182 and 3330. ICD-10-CM codes to identify atypical PD were G3183, G903, G239, G238, G232, G230, and G231. [a] Patients who met the criteria to enter the PDP cohort were evaluated at their psychosis diagnosis date and were matched; patients with PD without psychosis who were selected for the matched cohort were evaluated at the date of the matched PD diagnosis. [b] The number of events were assumed to follow a Poisson distribution. Therefore, corresponding exact 95% CIs were computed using methods described in Dobson et al. (1991). [c] The number of events were assumed to follow a Poisson distribution. Therefore, corresponding exact 95% CIs were computed using methods described in Sahai and Kurshid (1996).
(DOCX)

## Acknowledgments

The authors acknowledge the contribution of Shannon Hunter (RTI-HS) for review of statistical programming and Heather Danysh (RTI-HS) for coordination of the project, and Daniel Siepert and John Forbes (RTI-HS) for editorial assistance.

## Author Contributions

**Conceptualization:** Joan Forns, J. Bradley Layton, Mary Ellen Turner, Colleen Dempsey, Mary Anthony, Mary E. Ritchey, George Demos.

**Data curation:** Jennifer Bartsch.

**Formal analysis:** Jennifer Bartsch.

**Funding acquisition:** Mary Ellen Turner, Colleen Dempsey, Mary E. Ritchey.

**Investigation:** Joan Forns, J. Bradley Layton, Mary Anthony, Mary E. Ritchey.

**Methodology:** Joan Forns, J. Bradley Layton, Jennifer Bartsch, Mary Ellen Turner, Colleen Dempsey, Mary Anthony, Mary E. Ritchey.

**Project administration:** Joan Forns, J. Bradley Layton, Mary Anthony, Mary E. Ritchey.

**Resources:** Joan Forns, J. Bradley Layton, Mary Ellen Turner, Colleen Dempsey, Mary E. Ritchey.

**Software:** Jennifer Bartsch.

**Supervision:** Mary Ellen Turner, Mary Anthony, Mary E. Ritchey, George Demos.

**Writing – original draft:** Joan Forns.

**Writing – review & editing:** Joan Forns, J. Bradley Layton, Mary Ellen Turner, Colleen Dempsey, Mary Anthony, Mary E. Ritchey, George Demos.

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
