## [Decision Letter · Decision Letter 0]

28 Aug 2020

PONE-D-20-10989

Increased risk of falls and fractures in patients with psychosis and Parkinson disease

PLOS ONE

Dear Dr. Forms,

Thank you for submitting your manuscript to PLOS ONE. After careful consideration, we feel that it has merit but does not fully meet PLOS ONE’s publication criteria as it currently stands. Therefore, we invite you to submit a revised version of the manuscript that addresses all the points raised during the review process.

We look forward to receiving your revised manuscript.

Kind regards,

Gianluigi Forloni

Academic Editor

PLOS ONE

Journal Requirements:

2. We note that you have provided the following information in your Ethics Statement: "Institutional review board and/or any other required reviews of the

study protocol by specific committees were obtained in accordance with applicable national and local regulations.".Please revise this statement to indicate whether you obtained ethics approval to carry out the study. If you did obtain ethical approval to carry out the study please include the name of the committee that provided approval. If you did not obtain ethical approval please remove this fragment.

3.We note that you have indicated that data from this study are available upon request. PLOS only allows data to be available upon request if there are legal or ethical restrictions on sharing data publicly. For information on unacceptable data access restrictions, please see http://journals.plos.org/plosone/s/data-availability#loc-unacceptable-data-access-restrictions.

3.Thank you for stating the following in the Financial Disclosure section:

[This study was conducted by RTI Health Solutions under the direction of ACADIA Pharmaceuticals Inc and was funded by ACADIA Pharmaceuticals Inc.].   

We note that one or more of the authors are employed by a commercial company: RTI Health Solutions and ACADIA Pharmaceuticals Inc

Reviewers' comments:

Reviewer's Responses to Questions

**Comments to the Author**

1. Is the manuscript technically sound, and do the data support the conclusions?

Reviewer #1: Yes

Reviewer #2: Yes

2. Has the statistical analysis been performed appropriately and rigorously? 

Reviewer #1: Yes

Reviewer #2: Yes

3. Have the authors made all data underlying the findings in their manuscript fully available?

Reviewer #1: No

Reviewer #2: Yes

4. Is the manuscript presented in an intelligible fashion and written in standard English?

Reviewer #1: Yes

Reviewer #2: Yes

5. Review Comments to the Author

Reviewer #1: In this paper, a study is presented that examines if risk of falls and fractures differs between matched groups of people with Parkinson’s disease with and without psychosis. Large commercial insurance databases were used for this evaluation, which has some limitations as the authors mention in the discussion. The paper is well written. I have only some minor questions/suggestions:

•The term person-time needs to be explained.

•The criteria mentioned in lines 129-135 are exactly the same as presented in lines 102-109. See if this can be avoided to improve readability.

•People with PD before diagnosis of psychosis are considered to belong to the PD cohort. Thus, with the matching, one person can be in both groups (PD and PDP). How many people were in both groups? Does this influence the outcomes?

Reviewer #2: This study uses administrative claims to evaluate the incidence of falls and fractures between patients with Parkinson’s disease with and without psychosis. The manuscript describes methodology in good detail and the conclusions are supported by the data. The manuscript is well written and details ethical standards that were met.

This report finds an increased risk of falls and fractures among matched PD and PDP patients (IRR=1.44). These findings are supported by the clinical characteristics and challenges of patients with psychosis as well as findings in previous studies. While the methodology is rigorously detailed, including great efforts to properly match PD and PDP patients based on propensity scores, a few questions remain to fully explain the outcomes. Recommendations to address these concerns are as follows:

1. Was the possibility of atypical PD considered in selection of PD patients? It is suggested to specify the specific ICD codes used. If available data permit, it would be beneficial to exclude patients with a diagnosis (initially or at any follow up visit) of MSA, PSP, DLB, etc. to avoid diagnostic misclassification.

2. In Figure 1, it is unclear where “Patients with PD and PDP diagnosis on same day n=502” fit. Should there be an arrow connecting them to the initial block of “Patients with psychosis diagnosis n=16,955”? Or elsewhere?

3. Although the authors stated they matched by disease trajectory, this is rather limited using administrative data. There are unknown details about diagnostic delay, progression of clinical symptoms, etc. These limitations should be discussed. It would be more appropriate to describe the matching as being based on time since PD diagnosis rather than disease trajectory.

4. In the group of PDP patients, was psychosis present consistently after assignment to that group or is there a possibility of transient psychosis? This should be discussed, including potential implications of inclusion of transient psychosis, if applicable.

5. In the discussion, the authors raise the important points that onset of psychosis could be associated with comorbidities and that changes to antiparkinsonian medications could confound the association of PDP and falls. The authors also assert that psychosis appears to be an independent risk factor to explain the risk of falls. Although the authors addressed the potential for confounding through the use of a complex matching scheme, presenting an additional multivariate model to assess the risk of falls while controlling for disease duration, PD medication dose at time of fall, PD duration, and select comorbidities would further increase the confidence in the conclusion that psychosis is indeed an independent risk factor for falls. These results would also allow assessment of the relative contribution of each of the included factors in the risk of falls.

With these issues addressed, this manuscript offers valuable insight into a clinically important issue. The manuscript is very well written with excellently detailed methodology, organization and writing style.

6. PLOS authors have the option to publish the peer review history of their article (what does this mean?). If published, this will include your full peer review and any attached files.

Reviewer #1: No

Reviewer #2: No

---

## [Author Response · Author response to Decision Letter 0]

7 Jan 2021

Reviewer #1: In this paper, a study is presented that examines if risk of falls and fractures differs between matched groups of people with Parkinson’s disease with and without psychosis. Large commercial insurance databases were used for this evaluation, which has some limitations as the authors mention in the discussion. The paper is well written. I have only some minor questions/suggestions:

We appreciate the positive feedback provided.

The term person-time needs to be explained.

Person-time is the amount of observable time an individual or group contributes to follow-up. For example, one person followed for 2 years contributes 2 person-years; 10 people each followed for 6 months contribute a total of 5 person-years. We have added a clarification to page 6, line 108. Additionally, we have further clarified the term “immortal person-time,” which is mentioned on page 6, line 95: “(time during which a patient contributes to follow-up but has not yet met all eligibility criteria, and therefore any fall or fracture would be ineligible to be counted as an outcome).”

The criteria mentioned in lines 129-135 are exactly the same as presented in lines 102-109. See if this can be avoided to improve readability.

The exclusion criteria for the PDP cohort have been streamlined (page 7, line 123): “The same exclusion criteria applied at the time of PD cohort eligibility date, except the psychosis diagnosis, were applied again at the date of the psychosis diagnosis by using all available claims prior to the PDP index date.”

People with PD before diagnosis of psychosis are considered to belong to the PD cohort. Thus, with the matching, one person can be in both groups (PD and PDP). How many people were in both groups? Does this influence the outcomes?

Thank you for raising this good point. In the crude, unmatched population, essentially all patients in the PDP cohort were previously in the PD cohort, as shown in Figure 1. After matching, 9.2% of patients in the PD matched cohort were also in the PDP cohort. However, by design, having a fall/fracture while a patient is in the PD group, do not preclude from having a fall/fracture outcome in the PDP group. In other words, by construction, the patient’s two timepoints are independent. A patient was not allowed to match to their self. In addition, a patient was never in both cohorts at the same time. Once the patient had a psychosis diagnosis, the time in the PD cohort ended and the PDP cohort time started. So, the time periods are completely independent. Finally, when the patient entered the PDP cohort, we reevaluated their covariates at this time. So even though a patient could appear twice, we are treating them as separate patients (ie new covariates).

In summary, we consider that due to the reasons above mentioned, the fact that a patient can be in both groups had no meaningful impact of the conclusions of the study. 

 

Reviewer #2: This study uses administrative claims to evaluate the incidence of falls and fractures between patients with Parkinson’s disease with and without psychosis. The manuscript describes methodology in good detail and the conclusions are supported by the data. The manuscript is well written and details ethical standards that were met.

This report finds an increased risk of falls and fractures among matched PD and PDP patients (IRR=1.44). These findings are supported by the clinical characteristics and challenges of patients with psychosis as well as findings in previous studies. While the methodology is rigorously detailed, including great efforts to properly match PD and PDP patients based on propensity scores, a few questions remain to fully explain the outcomes.

We appreciate the positive feedback provided.

Recommendations to address these concerns are as follows:

1. Was the possibility of atypical PD considered in selection of PD patients? It is suggested to specify the specific ICD codes used. If available data permit, it would be beneficial to exclude patients with a diagnosis (initially or at any follow up visit) of MSA, PSP, DLB, etc. to avoid diagnostic misclassification.

Thank you for this comment. We have included additional details about the specific diagnosis codes used in our case definition of PD on page 5, lines 84-87. To increase the likelihood that those identified as having PD truly do have PD, our case definition required one inpatient claim for PD, two outpatient claims for PD separated by at least 30 days but within 365 days, or one outpatient claim for PD and at least two prescription claims for PD medication.

Additionally, as recommended by the reviewer, we evaluated the potential impact of patients with atypical PD. We calculated the percentage of patients with claims for atypical PD (Lewy body dementia or diseases of basal ganglia) at baseline. The number of patients in the matched and unmatched cohorts with these claims is included below: 

 Population Cohort Patients with Exclusion Percent

Unmatched PD cohort with atypical PD 154,339 3,371 2.2

Unmatched PDP cohort with atypical PD 12,132 1,393 11.5

Matched PD cohort with atypical PD 24,164 2,678 11.1

Matched PDP cohort with atypical PD 12,082 1,385 11.5

Based on the numbers above, we decided to include a sensitivity analysis excluding those patients with a diagnosis of atypical PD at baseline. Thus, we have described this in the methods section (page 11, line 209): “Finally, due to the possibility of PD diagnostic misclassification, we repeated the main analysis but excluded patients with a diagnosis code for atypical PD (i.e., dementia with Lewy body, multiple system atrophy, or progressive supranuclear palsy) at baseline.”

Results of this sensitivity analysis have been included as S4 table and S5 table and have been described as follows in the results section (page 19, line 314): “A total of 2,678 (11.1%) and 1,385 (11.5%) patients in the matched PD and PDP cohorts were excluded due to the co-occurrence of a diagnosis code for atypical PD (Lewy body dementia or degenerative diseases of the basal ganglia). After those exclusions, the characteristics of the remaining patients in the matched cohort were well-balanced between the PD without psychosis and PDP groups, as evidenced by the SMDs for all characteristics being near 0, between −0.05 and 0.03 (S4 Table). Comparative matched analyses excluding patients with atypical PD were almost identical to those observed in the main analysis (S5 Table); patients with PDP had higher IRRs of the composite falls/fractures outcome (IRR = 1.44; 95% CI, 1.39-1.50), falls (IRR = 1.48; 95% CI, 1.43-1.54), and fractures (IRR = 1.18; 95% CI, 1.08-1.28).”

Finally, we have added a description of this analysis in the discussion section (page 23, line 409): “The possibility of diagnostic misclassification due to the inclusion of patients with atypical PD cannot be fully rejected. Our strict PD case definition used in the present study reduced the risk of diagnostic misclassification. Moreover, main analyses were repeated excluding patients with atypical PD at baseline. The results of this sensitivity analysis were almost identical as observed in the main analysis.”

2. In Figure 1, it is unclear where “Patients with PD and PDP diagnosis on same day n=502” fit. Should there be an arrow connecting them to the initial block of “Patients with psychosis diagnosis n=16,955”? Or elsewhere?

Of the 154,481 patients with a PD diagnosis after inclusion and exclusion criteria, there were 502 patients who had a psychosis diagnosis on the PD cohort eligibility date. These 502 patients could not contribute to the PD cohort because of the psychosis diagnosis, but they did contribute to the PDP group because they met all the inclusion criteria for the PDP. However, these 502 patients are already included in the above box where the total of 154,481 patients are included. We have modified the text in the box “Patients with a psychosis diagnosis on the PD cohort eligibility date” and added a footnote to clarify any remaining question: “a Those 502 patients with a PD diagnosis and all eligibility criteria could not contribute time to the PD without psychosis group because they were diagnosed with psychosis on the cohort eligibility date.”

3. Although the authors stated they matched by disease trajectory, this is rather limited using administrative data. There are unknown details about diagnostic delay, progression of clinical symptoms, etc. These limitations should be discussed. It would be more appropriate to describe the matching as being based on time since PD diagnosis rather than disease trajectory.

We appreciate the reviewer’s comment, and we agree that the exact point in the disease trajectory is difficult to define using claims data alone. First, we have modified the terminology across the documents and now use “matched based on time since cohort entry and clinical characteristics.” Additionally, we have added a new limitation in the Discussion section (page 23, line 401): “We attempted to match patients in the PD and PDP cohorts at similar disease trajectories based on time since cohort entry (i.e., first noted PD diagnosis) and relevant clinical claims. Nevertheless, the success of matching patients at similar disease trajectories is unclear due to the lack of granular clinical data in administrative claims data, such as severity of initial symptoms, progression of clinical symptoms, and other key variables.”

4. In the group of PDP patients, was psychosis present consistently after assignment to that group or is there a possibility of transient psychosis? This should be discussed, including potential implications of inclusion of transient psychosis, if applicable.

Following the comment of the reviewer, we have added this as a limitation (page 23, line 405): “In the same line, the possibility that a patient with transient psychosis has been classified as PDP cannot be rejected. Our eligibility criteria in the PDP cohort only required the presence of a psychosis diagnosis after the PD diagnosis without requiring repeated codes. Claims data may lack the necessary granularity to define duration or intensity of symptoms.”

5. In the discussion, the authors raise the important points that onset of psychosis could be associated with comorbidities and that changes to antiparkinsonian medications could confound the association of PDP and falls. The authors also assert that psychosis appears to be an independent risk factor to explain the risk of falls. Although the authors addressed the potential for confounding through the use of a complex matching scheme, presenting an additional multivariate model to assess the risk of falls while controlling for disease duration, PD medication dose at time of fall, PD duration, and select comorbidities would further increase the confidence in the conclusion that psychosis is indeed an independent risk factor for falls. These results would also allow assessment of the relative contribution of each of the included factors in the risk of falls.

We appreciate the comment of the reviewer. The matching scheme used in our analysis did control for all baseline covariates. Since separate propensity score models were constructed in each time block, and all patients’ covariates were updated at each date and used in each block. All covariates were balanced after matching as shown in Supplemental Table S3, and therefore no confounding by the measured covariates should be expected.

The objective of the current study was to evaluate the incidence rates of falls and fractures in PDP patients compared with matched PD patients rather than to build a predictive model of the risk of falls and fractures. Thus, the purpose of the covariates is to control confounding between the PD and PDP groups, which we have demonstrated through the balance of the covariates. It would be inappropriate to adjust for factors measured after the beginning of follow-up. The reviewer’s suggestion of a separate generation of a multivariate predictive model is interesting, but it is incompatible with the study design we have implemented here. The contribution of individual risk factors measured at the time of the fall was not part of our study objectives.

With these issues addressed, this manuscript offers valuable insight into a clinically important issue. The manuscript is very well written with excellently detailed methodology, organization and writing style.

Thank you again for the positive feedback provided.

---

## [Editor Report · Decision Letter 1]

14 Jan 2021

Increased risk of falls and fractures in patients with psychosis and Parkinson disease

PONE-D-20-10989R1

Dear Dr. Forms,

We’re pleased to inform you that your manuscript has been judged scientifically suitable for publication and will be formally accepted for publication once it meets all outstanding technical requirements.

Kind regards,

Gianluigi Forloni

Academic Editor

PLOS ONE
---

## [Editor Report · Acceptance letter]

18 Jan 2021

PONE-D-20-10989R1 

Increased risk of falls and fractures in patients with psychosis and Parkinson disease 

Dear Dr. Forns:

I'm pleased to inform you that your manuscript has been deemed suitable for publication in PLOS ONE. Congratulations! Your manuscript is now with our production department. 

Kind regards, 

on behalf of

Dr. Gianluigi Forloni 

Academic Editor

PLOS ONE